# Rational Design of a Pan-Coronavirus Vaccine Based on Conserved CTL Epitopes

**DOI:** 10.3390/v13020333

**Published:** 2021-02-21

**Authors:** Minchao Li, Jinfeng Zeng, Ruiting Li, Ziyu Wen, Yanhui Cai, Jeffrey Wallin, Yuelong Shu, Xiangjun Du, Caijun Sun

**Affiliations:** 1School of Public Health (Shenzhen), Sun Yat-sen University, Shenzhen 518107, China; limch7@mail2.sysu.edu.cn (M.L.); zengjf7@mail2.sysu.edu.cn (J.Z.); lirt9@mail2.sysu.edu.cn (R.L.); wenzy3@mail2.sysu.edu.cn (Z.W.); shuylong@mail.sysu.edu.cn (Y.S.); 2Key Laboratory of Tropical Disease Control (Sun Yat-sen University), Ministry of Education, Guangzhou 514400, China; 3Gilead Sciences, Inc. 333 Lakeside Dr, Foster City, CA 94404, USA; Jenny.cai1@gilead.com (Y.C.); Jeffrey.Wallin@gilead.com (J.W.)

**Keywords:** coronavirus, pan-vaccine, antigen design, conserved epitopes

## Abstract

With the rapid global spread of the Coronavirus Disease 2019 (COVID-19) pandemic, a safe and effective vaccine against human coronaviruses (HCoVs) is believed to be a top priority in the field of public health. Due to the frequent outbreaks of different HCoVs, the development of a pan-HCoVs vaccine is of great value to biomedical science. The antigen design is a key prerequisite for vaccine efficacy, and we therefore developed a novel antigen with broad coverage based on the genetic algorithm of mosaic strategy. The designed antigen has a potentially broad coverage of conserved cytotoxic T lymphocyte (CTL) epitopes to the greatest extent, including the existing epitopes from all reported HCoV sequences (HCoV-NL63, HCoV-229E, HCoV-OC43, HCoV-HKU1, SARS-CoV, MERS-CoV, and SARS-CoV-2). This novel antigen is expected to induce strong CTL responses with broad coverage by targeting conserved epitopes against multiple coronaviruses.

## 1. Introduction

A novel highly pathogenic human coronavirus (HCoV) named severe acute respiratory syndrome coronavirus 2 (SARS-CoV-2) was found in December 2019. Soon thereafter, the rapid transmission of SARS-CoV-2 resulted in the global pandemic that remains a worldwide concern. There have been over 100 million case reports of Coronavirus Disease 2019 (COVID-19), and over 2 million death reports due to COVID-19 pandemic as of early February 2021 [1]. Coronavirus (CoV) is a single-stranded positive RNA virus that can be divided into four genera: α, β, γ, and δ. The genetic component of CoVs encodes four major structural proteins and sixteen nonstructural proteins (such as nsp1-nsp16) [2]. The structural proteins are thought to be the major targets for developing CoV vaccines, including spike-surface glycoprotein (S), small envelope protein (E), matrix protein (M), and the nucleocapsid protein (N).

Currently, seven HCoV species have been identified, including HCoV-NL63, HCoV-229E, HCoV-OC43, HCoV-HKU1, SARS-CoV, MERS-CoV, and SARS-CoV-2 [2]. There have been multiple outbreaks of various species of coronavirus recently. One hypothesis of the virus origin is the transmission from wildlife animals to humans (cross-species virus transmission). Human outbreak-associated coronavirus strains have included SARS-CoV (2002), MERS-CoV (2012), and SARS-CoV-2 (2019). The case death rate due to HCoV infection is 9.6% for SARS, 34.4% for MERS, and approximately 2.1% for COVID-19 [3]. Hence, an urgent research effort is needed to develop a safe and effective pan-CoV vaccine that can induce a broad cytotoxic T lymphocyte (CTL) response against different HCoV infections.

Thus, in this study, we investigated the conserved epitopes existed in all known HCoVs, and then developed a novel antigen based on a genetic algorithm of mosaic strategy, which have a broad coverage of CTL epitopes for known HCoVs. Our data support that there is a possibility for developing pan-coronavirus vaccines by targeting conserved CTL epitopes by the designed antigens.

## 2. Materials and Methods

### 2.1. Phylogenetic Analysis and Sequence Alignment for Coronavirus

All genomic sequences used in this study were collected from National Center for Biotechnology Information (NCBI), Virus Pathogen Resource (VIPR), and Global Initiative on Sharing All Influenza Data (GISAID), including 3,132 viral sequences (928 HCoV sequences and 2,204 other CoV sequences). The full-length genomes of representative HCoV and CoV viruses were analyzed by sequence alignment using MAFFT software (version 7.453). The auto strategy and phylogenetic analysis were performed FastTree (version 2.1.11) with GTR + CAT model.

### 2.2. Identification of Potential CTL Epitope

The CTL epitopes in the four structural proteins (S, M, N, E) of HCoVs were analyzed using the NetMHCpan-4.0 EL 4.0 algorithm and a size range of 9-mer amino acid (AA) [4]. The top 15 of mostly frequent human leukocyte antigens (HLA) class I alleles in the Chinese population (with frequency cutoff ≥ 6%), including HLA-A*02:01, HLA-A*24:02, HLA-A*11:01, HLA-A*33:03, HLA-A*30:01, HLA-B*40:01, HLA-B*46:01, HLA-B*13:02, HLA-C*04:01, HLA-C*03:03, HLA-C*08:01, HLA-C*03:04, HLA-C*06:02, HLA-C*01:02, and HLA-C*07:02 [5], were selected as a model to analyze the potential CTL epitopes. The frequency of each selected HLA allele in the Chinese population is shown in Table 1. Of note, these HLA alleles including HLA-A*11:01, HLA-A*24:02, HLA-A*02:01, HLA-C*07:02, HLA-C*06:02, and HLA-C*03:04 [6,7], are prevalent not only in Chinese population but also in other population worldwide (Appendix A). Therefore, our antigen design strategy is potentially applicable to other populations other than Chinese population.

We analyzed the potential CTL epitope distribution for four structural proteins with rank < 0.5 (most likely to be considered as high-affinity epitopes), based on the reference sequences of seven HCoVs species from the NCBI database (Table 2). Those peptides that can be potentially recognized by HLA multiple alleles were considered as components for synthesizing the following mosaic proteins.

### 2.3. Mosaic Antigen Design

We obtained 534 sequences of S protein, 485 sequences of M protein, 410 sequences of N protein, and 478 sequences of E protein with optimization to avoid redundancy in the dataset by using the CD-Hit tool to remove laboratory strains based on annotation information from NCBI with sequence similarity threshold at 0.9 and word length at 5 [8], These sequences were submitted to the Mosaic Vaccine Designer in *fasta* format using the following parameters: cocktail size at 4, epitope length at 9 AA, rare threshold at 3, run time at 10 h, population size at 200, cycle time at 10, stall time at 10, and internal crossover probability at 0.5. The average epitope with coverage between mosaic and natural viral proteins was determined by using the Epitope Coverage Assessment Tool (Epicover) with the following parameters: nominal epitope length at 9 AA, the maximum amino acid mismatches to score at 2, and the minimum number of occurrences of the potential epitope in viral protein set to 3 AA. The distribution of mosaic sequences in each viral species was analyzed by the Positional Epitope Coverage Assessment Tool (Posicover) with the following parameters: nominal epitope length at 9 AA, and antigen counts to compute upper bounds at 3,4.

### 2.4. Phylogenetic and Sequence Analysis of Mosaic Antigen Cocktail

The sequences of structural proteins involved in this study were aligned using MAFFT software (version 7.453) with auto strategy, and phylogenetic analysis was performed with FastTree (version 2.1.11) with GTR + CAT model. 

### 2.5. Computational Model of Three-Dimensional (3-D) for Mosaic Antigen

The sequences of mosaic proteins were submitted to the SWISS-MODEL and homology-derived conformational model were generated with 6NZK (from OC43), 6G13 (from MERS-CoV), 5C8S (from SARS-CoV), and 2MM4 (from SARS-CoV) as templates for S, N, M and E proteins, respectively. All models were uploaded to CHIMERA (version 1.14) for 3-D modeling. The degree of the structural similarity of mosaic antigen to the natural viral proteins was evaluated using the Qualitative Model Energy Analysis (QMEAN) method.

## 3. Results and Discussion

### 3.1. Phylogenetic Analysis of CoV Sequences in This Study

To facilitate the development of a pan-coronavirus vaccine, we aimed to design a novel antigen with potential to induce T cell responses to epitopes from various evolutionary coronaviruses. All sequences in this study were performed with phylogenetic analysis to determine their evolution relationship. Consistent with previous reports, our data indicated that these sequences could be divided into four clades (Figure 1). We focused on seven species of coronaviruses that can cause respiratory diseases in humans. Except for HCoV-229E and HCoV-NL63 belonging to the α-CoVs, the remaining 5 species belong to β-CoVs. Four species can cause mild upper respiratory disease, including HCoV-229E, HCoV-NL63, HCoV-HKU1, and HCoV-OC43. Meanwhile, MERS-CoV, SARS-CoV, and SARS-CoV-2 can infect the lower respiratory tract and cause severe respiratory syndrome in humans. In order to design an optimal antigen with broad coverage across multiple HCoV species, it is crucial to understand the genetic diversity and evolution link between all known HCoVs and other coronaviruses. As the major targets for developing vaccine, the structural proteins (S, E, M, and N) from 928 HCoV sequences were therefore selected for our vaccine target for further analysis.

### 3.2. Analysis of Potential CTL Epitopes with Existing HCoV Sequences

To investigate the potential epitopes that can be recognized to induce CTL responses, we analyzed the sequences of these structural proteins as described in Section 2.2. As shown in Table 3, the S protein had a high abundance of CTL epitopes than other three proteins. However, due to a shorter peptide length, the M protein had an even higher amount of CTL epitopes per 100 amino acids. The information of these potential epitopes was provided in detail in Appendix A. Our results were consistent with recent studies that the structural proteins (S, E, N, and M) contained extensive epitopes with the capability to induce strong CTL immune responses [9]. 

### 3.3. Design of the Mosaic CoV Antigens Based on Conserved CTL Epitopes

We analyzed the above sequences of S, M, N, and E proteins by using a genetic algorithm of mosaic strategy [10]. The rationale of the mosaic antigen design is to maximize the coverage of potential T-cell epitopes found in the circulating virus. The mosaic protein was comprised of short peptide fragments (each includes a length of 9 amino acid (AA) according to the common length of CTL epitope) identified from natural sequences (Figure 2A). Finally, we got four mosaic sequences as a cocktail set for each structural protein (Appendix A). The mosaic antigen has been shown to be promising for developing vaccines against highly variable pathogens, including HIV and influenza [11,12]. Based on a high-performance computational analysis, we identified a set of mosaic sequences for known HCoV S, M, N, and E proteins. 

We subsequently analyzed the potential CTL epitopes in these mosaic proteins. The Epitope Coverage Assessment Tool was used to calculate the proportion of CTL epitope in a set of circulating viral proteins sequences that are covered or matched by the CTL epitope in mosaic cocktail. The results are represented as mean epitope-coverage over the circulating viral proteins sequences. The reference sequences for S, N, M, and E proteins from seven HCoVs were listed in Table 2. A total of 928 HCoV sequences were included in our study. Each HCoV protein can have a different amount of sequence reported due to the availability of sequencing data. There were 534, 485, 410, and 478 sequences reported for S, M, N, and E proteins, respectively (Table 3). Each set of mosaic cocktails for S, N, M, and E proteins achieved epitope coverage between 87.77%-92.86% for perfect matches (9AA/ 9AA). If 8 AA of 9 AA matches were included (these imperfect CTL epitopes work well in most conditions), the epitope coverage was up to 93.39%-95.53% (Figure 2B and Table 4). For example, the potential epitopes in a set of mosaic N protein cocktail represented 91.96% coverage (9AA/9AA match) or 95.53% coverage (8AA/9AA match), compared to all predicated epitopes among the reported 410 sequences of natural N proteins. 

We next used the Positional Epitope Coverage Assessment Tool to calculate the proportion of CTL epitope in a set of aligned circulating viral proteins sequences that are covered or matched by the CTL epitope in mosaic cocktail. The results are represented as epitope-coverage over the position of circulating viral proteins sequences. The results showed that the CTL epitopes in the mosaic antigen matched well with their counterparts in the circulating viruses (Figure 3C). In addition, we excluded those rare epitopes (fewer than 3 times in all natural sequences) from mosaic antigens to avoid impairment of protective epitopes. These results suggested that the CTL epitopes in a set of mosaic antigens contained epitopes exclusively with high frequency in the circulating viruses.

We also determined the evolutionary relationship between the mosaic proteins and the circulating proteins. As shown in Figure 3D, phylogenetic analysis revealed that the mosaic antigen set for each kind of structural protein was evenly distributed in the four clades, suggesting their potential representativeness to the existing HCoVs.

### 3.4. 3-D Structure Modeling for Mosaic CoV Antigen

An ideal vaccine should be able to induce humoral and cellular immunity with a proper balance, therefore, the antigen is expected to maintain the conformation representing the circulating virus with both linear and spatial epitopes. We used 3-D structure modeling to understand the conformation of our candidate mosaic antigen. The QMEAN scores for candidate mosaic S, mosaic N, mosaic M, and mosaic E proteins are −0.97, 0.8, −4.32, and −2.82 respectively. A score > −4.0 suggests a model with high similarity. Our mosaic S protein, mosaic N protein, and mosaic E protein had scores > −4.0, supporting that their 3-D structures were highly similar to their counterparts in circulating virus and likely retained their capability to induce humoral immune responses (Figure 3). 

## 4. Discussion

The vaccine efficacy can be affected by the antigen design, delivery vehicle, routine of administration, and adjuvant. Among them, antigen design is crucial for an effective vaccine. In this study, for developing a pan-HCoV vaccine, we therefore designed a novel mosaic antigen with a broad coverage by targeting conserved CTL epitopes of all the existing HCoV sequences.

A variety of HCoV vaccine strategies have been developing, including inactivated, attenuated live, subunit, DNA, mRNA, and recombinant vector vaccines. Among them, several kinds of SARS-CoV-2 vaccines have been evaluated in phase III clinical trials [13,14], and are becoming clinically available. The induction of neutralizing antibodies against S protein and its receptor-binding domain (RBD) has been extensively investigated in the design of SARS-CoV, MERS-CoV, and SARS-CoV-2 vaccines [15]. However, some studies have shown that S antigen-based vaccine candidates might cause detrimental immunopathology and aggravate disease progression by an antibody-dependent enhancement of infection (ADE) [16,17]. Alternatively, antigen-specific CTL responses are also crucial for controlling viral infection and replication [18]. In addition, one prominent feature of the SARS-CoV-2 infection is lymphopenia, which is mainly manifested with the decreased and exhausted natural killer cells and T cells, implying that a poor T cell response might be associated with the progression of severe COVID-19 patients. Therefore, the importance of T cell-mediated antiviral immunity should be further studied in the development of HCoV vaccines. 

A previous study showed that induction of airway memory T cell responses against N protein has led to the protection of multiple HCoV species, including SARS-CoV, MERS-CoV, and related bat CoV [19]. Data obtained from a recent study showed that long-lasting memory T cells could potentially respond to the N protein of SARS-CoV-2 in some infected individuals recovered from SARS-CoV infection more than 17 years ago [20]. Also, the SARS-CoV-2-specific CD4+ T lymphocytes cells were found in many unexposed healthy individuals, suggesting that there might be broadly cross-reactive T cells that can recognize SARS-CoV-2 and the circulating “common cold” human coronaviruses (such as HCoV-HKU1, HCoV-229E) [21]. Therefore, it is important to investigate the possibility of pan-HCoV vaccines by targeting broadly cross-reactive CTL epitopes.

The immune epitopes in this study were predicted using computational design, so there remained a concern if these epitopes would match to CTLs identified from the experimental studies. Indeed, the experimentally verified T cell epitopes reported in the Immune Epitope Database (IEDB) were mostly included in our predicted epitopes. For example, S protein and N protein-reactive CD8+ epitopes were increasingly identified from recovered COVID-19 patients by experimental methods [22,23], including S-34 (EYVSQPFLM), S-106 (KSTNLVKNK), NP-1 (QRNAPRITF), NP-16 (SPRWYFYYL, YLGTGPEAGL), and NP-51 (KTFPPTEPK). In addition, some epitopes targeted to M protein (PKEITVATSRTLSYYKL, FLWLLWPVTL, FLFLTWICL, KLLEQWNL) and E protein (FLLVTLAIL, SEETGTLIVNSVLLF) were also reported in recent works [22,23,24]. The result are shown in Table 5. Importantly, the above mentioned CD8+ T cell epitopes were all contained in our mosaic antigen sets, suggesting that the predicated epitopes in this work potentially match to the experimentally identified epitopes.

The mosaic proteins in our antigen were predicted to have 3-D structures resembling their counterparts found in circulating virus. There have been many reported data on the structure of CoVs S and N proteins (such as 6lvn, 6m1v, 6zgg, 6zp2 for S, and 6m3m, 6vyo, 6wji, 6wkp for N), so it is easier to match and construct the 3-D structure for our mosaic S and mosaic N proteins. However, the structures for the transmembrane domain of M and E proteins remain to be elucidated. As previously reported, it is challenging to predict the protein structure of a small transmembrane domain [25,26]. We had to use remote homologues (5c8s for M protein, and 2mm4 for E protein, respectively) for the 3-D conformational modeling with a low sequence similarity. If there are available high-resolution structure data for M and E protein in the future, our mosaic M and E proteins might exhibit good similarity and higher QMEAN scores. As a result, with the retention of linear epitopes and spatial epitopes, our mosaic antigen is expected to induce both humoral and cellular immune responses.

There are limitations in our study. The experimental data are needed to further verify this strategy, especially to test the in vivo immunogenicity and protective efficacy of our mosaic antigen-based pan-HCoV vaccine. Overall, our study provides a basis for advancing T cell-based pan-coronavirus vaccines with a rational design.

## Figures and Tables

**Figure 1 viruses-13-00333-f001:**
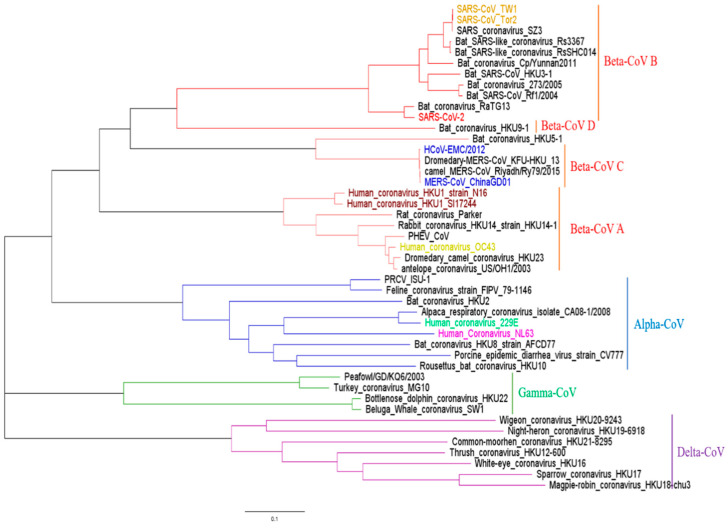
Phylogenetic analysis of full-length genomes of representative viruses of the existing coronavirus. Lines with different colors represent different coronavirus species, Alpha-coronaviruses (blue lines), Beta-coronaviruses (red lines), Gamma-coronaviruses (green lines), and Delta-coronaviruses (purple lines). Human isolates are highlighted with different colors, whereas strains from other hosts are shown in black. green: HCoV-229E; purple: HCoV-NL63; orange: SARS-CoV; red: SARS-CoV-2; blue: MERS-CoV; yellow: HCoV-OC43; brown: HCoV-HKU1. The scale bar is used to measure the branch length.

**Figure 2 viruses-13-00333-f002:**
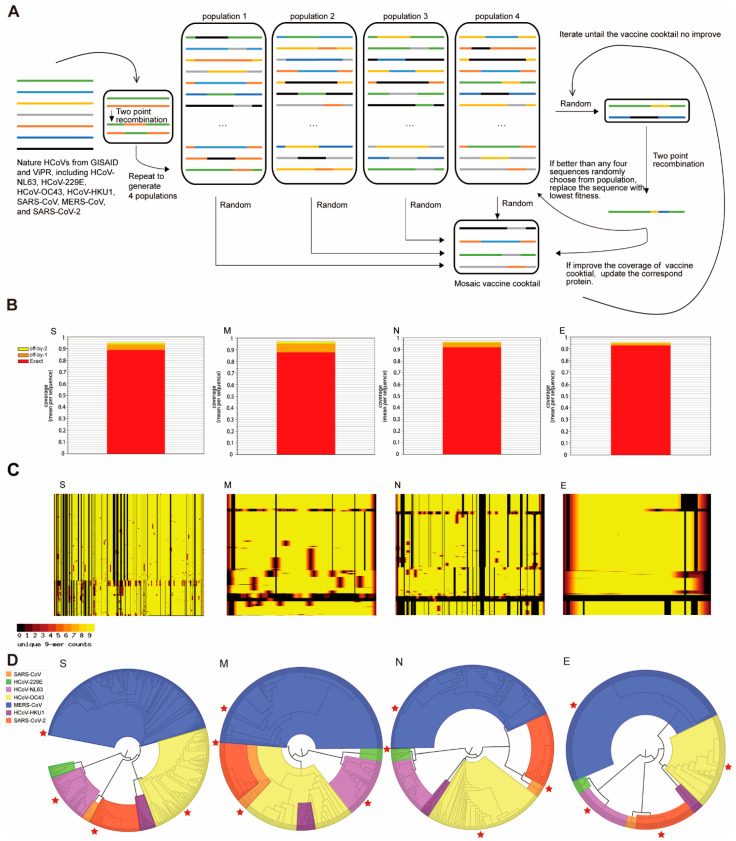
Design of mosaic antigen for a pan-coronavirus vaccine based on conserved CTL epitopes. (**A**) The schema of the genetic algorithm for the generation of a mosaic vaccine from seven species of HCoVs. (**B**) The mean epitope-coverage over the circulating viral proteins sequences covered by the CTL epitope in mosaic cocktail. The results indicated coverage (mean per-sequence) of S protein, M protein, N protein, and E protein. ‘Exact’ represents a 100% match, 9 AA out of 9 AA; ‘off-by-1’ indicates a match at 8AA out of 9 AA; ‘off-by-2’ indicates a match at 7AA out of 9 AA. (**C**) The alignment of the epitopes in mosaic antigens with their counterparts in proteins found in circulating virus using the Positional Epitope Coverage Assessment Tool. Each colored square includes an alignment of amino acid. Each “row” represents an amino acid sequence of the protein in the block. The column represents the relative position of a specific epitope. If a 9-aa in mosaic antigen makes a 100% match to their counterpart found in the vaccine or circulating virus from existing reports, the score is 9 and colored with light yellow; for a 9-aa without any matches, the score is 0 and colored with black. The scores in between are colored codes as the darkness increase. (**D**) Phylogenetic tree analysis of the mosaic antigens. The trees were midpoint rooted, and these sequences formed four clades, with HCoV-229E and HCoV-NL63 in clade 1; SARS-CoV and SARS-CoV-2 in clade 2; HCoV-HKU1 and HCoV-OC43 in clade 3; and the MERS-CoV in clade 4. Results showed that the mosaic S protein, M protein, N protein, and E protein were evenly distributed in the four clades. The red starts represent the genomic location of each mosaic protein. S: S protein; M: M protein; N: N protein; E: E protein.

**Figure 3 viruses-13-00333-f003:**
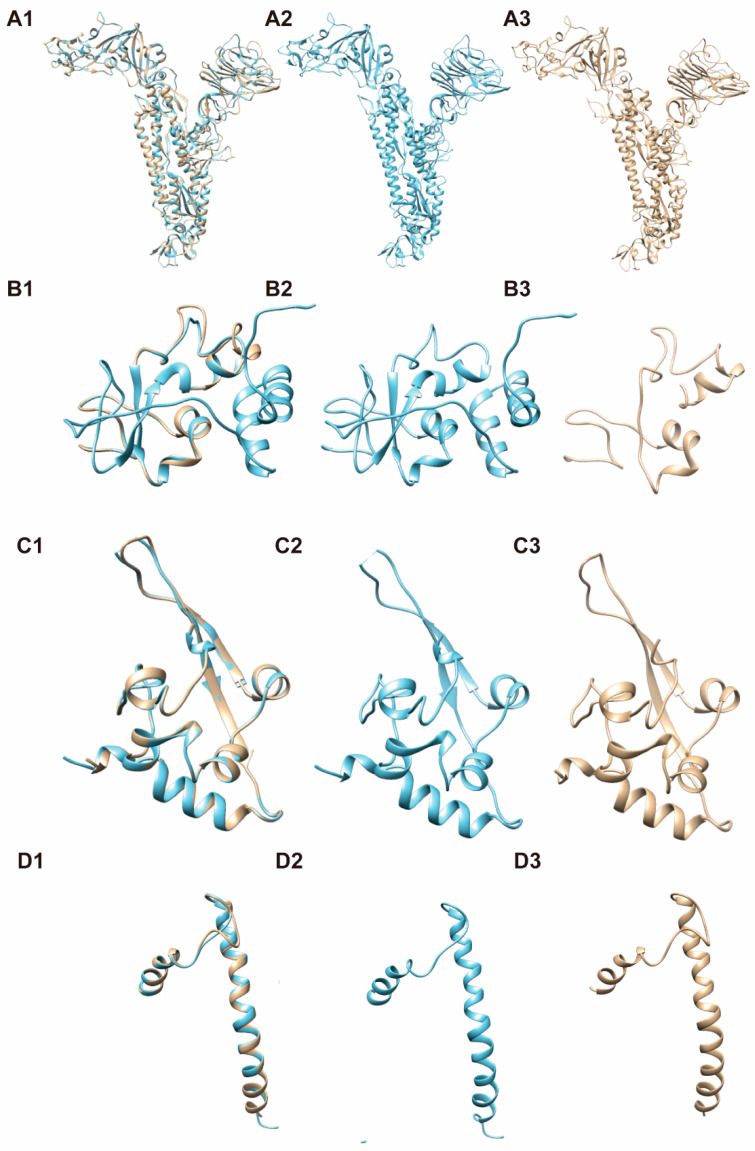
The three-dimensional structure models for these candidate mosaic proteins, based on the computer-guided homology modeling method. Mosaic sequences were submitted to the SWISS-MODEL server to construct the protein structure. We used PyMOL software to visualize the model, and then evaluated its quality with QMEAN tool. **A**, S protein; **B**, N protein; **C**, M protein; **D**, E protein; **1**, Merged images of mosaic and nature proteins; **2**, Proteins found in circulating virus (with PDB ID: 6NZK (from OC43), 6G13 (from MERS-CoV), 5C8S (from SARS-CoV) and 2MM4 (from SARS-CoV) respectively); **3**, Mosaic protein.

**Table 1 viruses-13-00333-t001:** The frequency of human leukocyte antigens (HLA) alleles among the Chinese population.

HLA Alleles	Frequency in Chinese Population
HLA-A*11:01	18.57%
HLA-A*24:02	15.73%
HLA-A*02:01	13.79%
HLA-A*33:03	8.03%
HLA-A*30:01	6.67%
HLA-B*40:01	9.59%
HLA-B*46:01	7.83%
HLA-B*13:02	7.11%
HLA-C*07:02	14.46%
HLA-C*01:02	14.10%
HLA-C*06:02	10.50%
HLA-C*03:04	8.90%
HLA-C*08:01	8.23%
HLA-C*03:03	7.90%
HLA-C*04:01	6.02%

**Table 2 viruses-13-00333-t002:** The Accession number of reference protein sequence from HCoVs.

Virus Name	S Protein	M Protein	N Protein	E Protein
SARS-CoV-2	YP_009724390.1	YP_009724393.1	YP_009724397.2	YP_009724392.1
MERS-CoV	YP_009047204.1	YP_009047210.1	YP_009047211.1	YP_009047209.1
SARS-CoV	NP_828851.1	NP_828855.1	NP_828858.1	NP_828854.1
HCoV-229E	NP_073551.1	NP_073555.1	NP_073556.1	NP_073554.1
HCoV-OC43	YP_009555241.1	YP_009555244.1	YP_009555245.1	YP_009555243.1
HCoV-NL63	YP_173238.1	YP_173241.1	YP_173242.1	YP_173240.1
HCoV-HKU1	YP_003767.1	YP_003770.1	YP_003771.1	YP_003769.1

**Table 3 viruses-13-00333-t003:** The number of potential cytotoxic T lymphocyte (CTL) epitopes in HCoVs structure proteins.

Name	S Protein			M Protein			N Protein			E Protein		
	HLA-A	HLA-B	HLA-C	HLA-A	HLA-B	HLA-C	HLA-A	HLA-B	HLA-C	HLA-A	HLA-B	HLA-C
SARS-CoV-2	105	88	107	34	15	27	35	16	21	11	6	6
MERS-CoV	116	101	121	24	21	32	32	17	16	8	7	5
SARS-CoV	108	84	111	36	20	26	35	16	21	9	8	6
HCoV-229E	104	95	111	30	24	28	32	21	26	8	4	7
HCoV-OC43	112	91	115	35	19	30	35	24	23	11	8	11
HCoV-NL63	125	110	133	29	15	21	30	19	30	12	5	13
HCoV-HKU1	122	97	129	28	21	25	30	20	29	13	6	11
Mosaic1	122	93	120	26	23	33	32	17	16	11	8	12
Mosaic2	105	88	107	29	15	20	34	22	27	8	7	5
Mosaic3	123	111	134	30	16	32	35	16	21	12	5	13
Mosaic4	117	101	121	37	15	24	31	23	27	11	6	6

**Table 4 viruses-13-00333-t004:** The mean epitope-coverage over the circulating viral proteins sequences covered by the CTL epitope in mosaic cocktail.

Mosaic Antigen Set (*n* = 4)	Subset Counts	Exact Match	Off-By-1 Match
S protein	534	0.8901	0.9339
M protein	485	0.8777	0.9505
N protein	410	0.9196	0.9553
E protein	478	0.9286	0.9447

**Table 5 viruses-13-00333-t005:** The epitope verified by experiment contained in the mosaic sequence.

Protein Name	Epitope Verified by Experiment	CD4+/CD8+ T Cell Response	HLA Restriction(s)	Position	Mosaic Sequence
S protein	EYVSQPFLM [22]	CD8+	A*11:01; C*07:02	169–177	Seq2
	KSTNLVKNK [22]	CD8+	A*02:01; B*40:01	529–537	Seq2
N protein	QRNAPRITF [22]	CD8+	A*11:01; B*40:01; C*07:02	9–17	Seq3
	SPRWYFYYL [22]	CD8+	A*02:01; C*07:02	105–113	Seq3
	YLGTGPEAGL [23]	CD8+	A*02:01; C*07:02	112–121	Seq3
	KTFPPTEPK [22]	CD8+	A*11:01; C*07:02	361–369	Seq3
M protein	PKEITVATSRTLSYYKL [22]	CD4+	DRB1*07:01;	164–180	Seq1
	FLWLLWPVTL [23]	CD8+	A*02:01	53–62	Seq4
	FLFLTWICL [23]	CD8+	A*02:01	26–34	Seq4
	KLLEQWNL [23]	CD8+	A*02:01	15–22	Seq4
E protein	FLLVTLAIL [23]	CD8+	A*02:01	26–34	Seq4
	SEETGTLIVNSVLLF [24]	CD4+	DQA1*0501; DQB1*0301	6–20	Seq4

## Data Availability

The data presented in this study are available in the main text and supplementary material.

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
