# Peer review of "Rational Design of a Pan-Coronavirus Vaccine Based on Conserved CTL Epitopes"

_viruses, 2021, doi:10.3390/v13020333_

Round 1
Reviewer 1 Report
In their manuscript, Li and co-authors describe an interesting approach for designing a pan-coronavirus vaccine which should induce protective immunity against all coronaviruses which can infect humans. The authors performed bioinformatic analyses of four CoV structural genes (S, M, N, and E) to design mosaic proteins with maximal CTL epitope coverage for all studied coronaviruses. In general, this approach is interesting, but there are a number of critical issues that need to be addressed prior to publication.
Major comments:
- The author claim that they design a CTL-based immunogen which can be further tested in wet labs. However, the most important information on the final epitope content in the suggested mosaic sequences is not present in the main text. It is very difficult to understand whether the most immunogenic confirmed CTL epitopes are preserved in these constructs or not. Supplementary materials have just raw data which is difficult to read.
- It is not clear from the text that there are four mosaic sequences for each structural protein which are combine in the cocktail (shown in supplementary material).
- It is known that CD4 T-cell epitopes are also important for induction of protective immunity (both humoral and cellular), but they were not analyzed in the manuscript. For example, the authors refer to a study about cross-protective role of lung-resident memory T cells (lanes 241-243), but that paper discussed CD4 T cells, not CTL.
- While designing a CTL-based vaccine, the authors assess conformational stability of the mosaic proteins and they rely only on the visual uniformity of the 3-D structures of real viral proteins and the mosaic constructs. However, they had to analyze the stability of B-cell epitopes as well, both linear and spatial. In addition, there need to be a guarantee that there are no epitopes with known ADE effect (which were shown for SARS virus before). No such analyze were performed.
Minor comments:
- Lane 43. Please define “CTL” when first used in the text.
- Lanes 95 and 219: please indicate to which CoV these structures are referred to.
Author Response
Major comments:
- The author claim that they design a CTL-based immunogen which can be further tested in wet labs. However, the most important information on the final epitope content in the suggested mosaic sequences is not present in the main text. It is very difficult to understand whether the most immunogenic confirmed CTL epitopes are preserved in these constructs or not. Supplementary materials have just raw data which is difficult to read.
Answer: Thank you for your questions. We have discussed in depth the recently published findings about relevant work, and cited those experimentally verified epitopes to prove that our mosaic antigen retained the T cell epitopes in the natural sequence. The epitope verified by experiment contained in mosaic sequence is showed in Table 5. We have added the following table in the corresponding paragraph to make our results more readable as you kindly suggested.
Table 5. The epitope verified by experiment contained in mosaic sequence
|
Protein name |
Epitope verified by experiment |
CD4+/CD8+ T cell response |
position |
Mosaic sequence |
|
S protein |
EYVSQPFLM |
CD8+ |
169-177 |
Seq2 |
|
KSTNLVKNK |
CD8+ |
529-537 |
Seq2 |
|
|
N protein |
QRNAPRITF |
CD8+ |
9-17 |
Seq3 |
|
SPRWYFYYL |
CD8+ |
105-113 |
Seq3 |
|
|
YLGTGPEAGL |
CD8+ |
112-121 |
Seq3 |
|
|
KTFPPTEPK |
CD8+ |
361-369 |
Seq3 |
|
|
M protein |
PKEITVATSRTLSYYKL |
CD4+ |
164-180 |
Seq1 |
|
FLWLLWPVTL |
CD8+ |
53-62 |
Seq4 |
|
|
FLFLTWICL |
CD8+ |
26-34 |
Seq4 |
|
|
KLLEQWNL |
CD8+ |
15-22 |
Seq4 |
|
|
E protein |
FLLVTLAIL |
CD8+ |
26-34 |
Seq4 |
|
SEETGTLIVNSVLLF |
CD4+ |
6-20 |
Seq4 |
- It is not clear from the text that there are four mosaic sequences for each structural protein which are combine in the cocktail (shown in supplementary material).
Answer: Thank you for your mention. We have added a description to show that there are four mosaic sequences for each structural protein in the revised Result 3.3.
- It is known that CD4 T-cell epitopes are also important for induction of protective immunity (both humoral and cellular), but they were not analyzed in the manuscript. For example, the authors refer to a study about cross-protective role of lung-resident memory T cells (lanes 241-243), but that paper discussed CD4 T cells, not CTL.
Answer: Thank you for your mentions. We agree with you that CD4 T-cell epitopes are also important for induction of protective immunity. Actually, recent studies showed that both CD4+ and CD8 + T cell responses were induced by SARS-CoV-2 infection, and these T cell responses are all important to mediate the broadly protective immunity (cross protection) against different kinds of coronavirus. As a proof-of- concept study, we mainly focus on the conserved CTL epitopes in this study.However, some experimentally verified CD4+ T epitopes were actually existed in our designed mosaic sequence (refer to question 1, and Table 5), so our strategy is expected to effectively induce both CD4+ and CD8 + T cell responses. As you suggested, we will expand this mosaic strategy to mainly analyze the sequences based on CD4+T cell epitopes in another project in future. We have modified our manuscript accordingly. Thank you for your understanding and suggestions.
- While designing a CTL-based vaccine, the authors assess conformational stability of the mosaic proteins and they rely only on the visual uniformity of the 3-D structures of real viral proteins and the mosaic constructs. However, they had to analyze the stability of B-cell epitopes as well, both linear and spatial. In addition, there need to be a guarantee that there are no epitopes with known ADE effect (which were shown for SARS virus before). No such analyze were performed.
Answer: Thanks for your comments. As you kindly said, as a proof-of- concept study, we mainly focus on designing a CTL epitopes-based vaccine in this study, and we developed a novel antigen to have a broad coverage of CTL epitopes for pan-coronavirus vaccines. Meantime, we also found that some experimentally verified CD4+ T epitopes were actually existed in our designed mosaic sequence. Furthermore, we constructed the 3-D structure modeling to understand the conformation of our candidate mosaic antigen, results showed that there were a highly similarity between mosaic proteins and their counterparts in circulating virus, which implied their capability to induce B-cell epitopes (both linear and spatial)-mediated humoral immune responses. However, prediction of B-cell epitopes (including ADE epitopes) is still a problematic scientific issue for this field. We will explore this mosaic strategy to analyze B cell epitopes in another challenging project in future. We have modified our manuscript accordingly. Thank you for your understanding and suggestions.
Minor comments:
- Lane 43. Please define “CTL” when first used in the text.
- Lanes 95 and 219: please indicate to which CoV these structures are referred to.
Answer to 5 and 6: Thank you for your careful reading. We have provided these information in the revised manuscript.
Reviewer 2 Report
The manuscript “viruses-1091148” entitled “Rational Design of a Pan-Coronavirus Vaccine Based on Conserved CTL Epitopes" is a well-written article. The authors could illustrate theoretically the design of a novel antigen with broad coverage based on the genetic algorithm of mosaic strategy. This antigen is expected to have a potentially broad coverage of conserved cytotoxic T lymphocyte (CTL) epitopes to the greatest extent, including the existing epitopes from all reported HCoV sequences (HCoV-NL63, HCoV-229E, HCoV-OC43, HCoV-HKU1, SARS-CoV, MERS-CoV, and SARS-CoV-2). This novel antigen is expected also to induce strong CTL responses with broad coverage by targeting conserved epitopes against multiple coronaviruses. However major points have to be done before publication:
- The immunogenicity and vaccine efficacy of the mosaic antigen is not experimentally validated. The authors should provide experimental data.
- Authors mixes in whole manuscript between MERS and SARS (disease) and MERS-CoV and SARS-CoV (Virus) (e.g. lines 36, 41, table 2).
- The quality of the figures is not satisfactory.
- There are some typos and scientific language inaccuracies in the whole manuscript. I would suggest the author to carefully review and amend it accordingly.
Author Response
The manuscript “viruses-1091148” entitled “Rational Design of a Pan-Coronavirus Vaccine Based on Conserved CTL Epitopes" is a well-written article. The authors could illustrate theoretically the design of a novel antigen with broad coverage based on the genetic algorithm of mosaic strategy. This antigen is expected to have a potentially broad coverage of conserved cytotoxic T lymphocyte (CTL) epitopes to the greatest extent, including the existing epitopes from all reported HCoV sequences (HCoV-NL63, HCoV-229E, HCoV-OC43, HCoV-HKU1, SARS-CoV, MERS-CoV, and SARS-CoV-2). This novel antigen is expected also to induce strong CTL responses with broad coverage by targeting conserved epitopes against multiple coronaviruses. However major points have to be done before publication:
(Ⅰ) The immunogenicity and vaccine efficacy of the mosaic antigen is not experimentally validated. The authors should provide experimental data.
Answer: Thanks for your valuable comments. As you mentioned, the experimental data are needed to further verify our strategy, especially the in vivo immunogenicity and protective efficacy for our mosaic antigen-based pan-HCoV vaccine. We totally agree with you that it will be a more important paper if we have more experimental data for this strategy. However, our current goal in this stage is to design a kind of novel antigen based on a genetic algorithm, which to the greatest extent has a broad coverage of conserved CTL epitopes from all known HCoV sequences, and provide theoretical basis to develop pan-coronavirus vaccines. Actually, our designed mosaic sequences are to contain mostly CTL epitopes from the natural sequences, especially the epitopes which are shared with various coronaviruses to elicit cross immunity. Of note, these epitopes matched well with the mostly published experimental data (please refer to Table 5 in revised manuscript). Meantime, we also found that some experimentally verified CD4+ T epitopes were actually existed in our designed mosaic sequence. Furthermore, we constructed the 3-D structure modeling to understand the conformation of our candidate mosaic antigen, results showed that there were a highly similarity between mosaic proteins and their counterparts in circulating virus, which implied their capability to induce B-cell epitopes (both linear and spatial)-mediated humoral immune responses. Based on present data, it warranted the next work to verify the immunogenicity and protection efficacy by animal experiments. However, the related laboratory experiments are time-consuming, especially to require the high-level biosafety laboratories (P3 level) to manipulate the related live viruses (SARS-CoV-2). We are looking for partners to carry out the relevant work, and we expect to perform it when conditions are permitted soon. We have mentioned these information and limitation in our revised manuscript. Thanks for your understanding and kind suggestions.
(Ⅱ) Authors mixes in whole manuscript between MERS and SARS (disease) and MERS-CoV and SARS-CoV (Virus) (e.g. lines 36, 41, table 2).
Answer: Thank you for your corrections. We have modified these mistakes carefully in the revised text.
(Ⅲ) The quality of the figures is not satisfactory.
Answer: Thank you for your kind mention. We have provided a higher resolution pictures for Fig2. and Fig3.
(Ⅳ) There are some typos and scientific language inaccuracies in the whole manuscript. I would suggest the author to carefully review and amend it accordingly.
Answer: Thank you for your careful reading. We have revised the English language carefully. These revisions were clearly highlighted using the "Track Changes" function in revised text. Thank you.
At last, we appreciate your valuable suggestions and comments regarding our manuscript, which are very helpful for improving our manuscript. We have taken into account of these comments seriously, and have made revisions in the manuscript accordingly. We look forward to hearing your feedback soon.
Sincerely,
Caijun Sun Ph.D
School of Public Health (Shenzhen), Sun Yat-sen University, Guangdong, China
Email: suncaijun@mail.sysu.edu.cn
Reviewer 3 Report
Manuscript Title:
Rational Design of a Pan-Coronavirus Vaccine Based on Conserved CTL Epitopes
Reference:
Viruses-1091148
Authors:
Minchao Li,, Jinfeng Zeng, Ruiting Li, Ziyu Wen, Yanhui Cai, Jeffrey Wallin,Yuelong Shu, Xiangjun Du, Caijun Sun.
Comments of the reviewer
The present manuscript, submitted by the groups led by Du and Sun, summarises the efforts intended to analyse the epitopes found in different HCoVs reported to date referred to the four structural proteins (S, M, N and E) of these pathogens. Based on the genetic algorithm of mosaic strategy, antigen sequences are proposed to induce strong CTL responses directed ultimately to help the development of pan-HCoV vaccines with broad coverage against several HCoVs.
After a critical reading, as a whole, the manuscript seems to this reviewer very well organised and data presented support a further exploration of the proposed antigens as promising antigens for the development broad spectrum HCoVs vaccines (the “proof-of-concept” of this piece of work). However, although still remains pending this task, the manuscript provides a convincing starting point as the rationally designed epitopes merges all the prerequisites to provide immune response as match the 3D structures of the experimentally-identified HCoVs immunogenic epitopes (there is a good correlation between the highlighted epitopes and the elucidated structure of the naive proteins).
The writing style is brief, concise and direct, with no concessions to excessive “decoration” on the description of the methods and results, but no needed data is missed. This is a merit as the whole text is straightforward and easily readable. The information required and tools to process it have been properly used. The research has been properly designed and correctly performed, whereas the data adequately treated and supports the conclusions. No concerns regarding the English language (only minor syntax or typographic error detected in the text). The introduction thematically locates the subject under study whereas the manuscript is adequately supported by up-to-date bibliographic references.
Globally, the organization of the contents, the structure, correct writing style and proper description and discussion of the results (pointed to reasonable conclusions) fits the minimum requirements for acceptability. Moreover, this work is focused on the development of responses against HCoVs, among them COVID19, the most concerning contemporary pandemic, with dramatic implications at health, social and economic levels. As a consequence, results described could be interesting for the specialised reader and valuable starting point for the eventual development of therapeutic tools in the fight against HCoVs, which deserve additional support.
Specific comments
1-. Do you plan any modification of the proposed sequences in order to improve/stabilise the spatial distribution of the selected epitopes? (change of some selected aa by modified aas, use of elements to lock-restrict conformation (disulphide bridges, etc.).
Page 1. Line 28-29
As the accessing date is 1st Dec 2020, when authors say “…COVID-19 pandemic as of Mid-January 2021 [1]” it should be more accurately “… late Dec 2020)
Page 2; line 60
Please, specify the meaning of the HLA acronym (Human leukocyte antigens)
Page 9; line 229
Is reference 14 properly cited?
Page 9; lines 230-231
“…investigated in the vaccine design of SARS-CoV, MERS-CoV, and SARS-CoV-2 vaccines [15].” One “vaccine” has to be removed (redundant)
Figures
Please, provide a higher resolution pictures/images used in all Figures (particularly for Fig. 2 and 3).
Summary of reviewer
This article fits the minimum quality required to be acceptable for publication in the current form.
Author Response
- Do you plan any modification of the proposed sequences in order to improve/stabilise the spatial distribution of the selected epitopes? (change of some selected aa by modified aas, use of elements to lock-restrict conformation (disulphide bridges, etc.).
Answer: Thank you for your valuable suggestions. As you mentioned, structure-based epitope design is an interesting field for antigen design to stabilize the spatial epitopes, and this kind of research is out of scope for this article, but we will consider it seriously in the further studies.
(2) Page 1. Line 28-29
As the accessing date is 1st Dec 2020, when authors say “…COVID-19 pandemic as of Mid-January 2021 [1]” it should be more accurately “… late Dec 2020)
Page 2; line 60
Please, specify the meaning of the HLA acronym (Human leukocyte antigens)
Page 9; line 229
Is reference 14 properly cited?
Page 9; lines 230-231
“…investigated in the vaccine design of SARS-CoV, MERS-CoV, and SARS-CoV-2 vaccines [15].” One “vaccine” has to be removed (redundant)
Answer: Thank you for your careful reading. We have corrected these mistakes in the corresponding paragraph in our revised text.
(3) Figures
Please, provide a higher resolution pictures/images used in all Figures (particularly for Fig. 2 and 3).
Answer: Thank you for your kind reminders. We have provided a higher resolution pictures for Fig2. and Fig3.
Round 2
Reviewer 1 Report
The authors properly adjusted their manuscript according to the reviewers’ critique. I only have several comments that need to be addressed prior to publication:
- Table 1. Either the authors should indicate that their vaccine is designed for Chinese population only, or this table should include other populations. The main text, not supplementary material, should provide information on HLA coverage.
- Table 5 was added in response to question about experimentally confirmed epitopes. However, this table should also contain information on HLA alleles and refer to studies where each epitope was investigated.
- Supplementary files should have their names and be indicated at the end of the manuscript (lanes 304-305).
Author Response
- Table 1. Either the authors should indicate that their vaccine is designed for Chinese population only, or this table should include other populations. The main text, not supplementary material, should provide information on HLA coverage.
Answer: Thanks for your kind mentions. As you mentioned, the frequency of HLA alleles varies in different population because of the genetic complexity in HLA region. In this study, we selected top 15 of mostly frequent HLA class I alleles in the Chinese population (frequency cutoff ≥ 6%) as a model to analyze the potential CTL epitopes. Of note, these HLA alleles including HLA-A*11:01, HLA-A*24:02, HLA-A*02:01, HLA-C*07:02, HLA-C*06:02 and HLA-C*03:04, are prevalent not only in Chinese population but also in other population worldwide. Therefore, our antigen design strategy is potentially applicable to other populations other than Chinese population. We have mentioned this information in our revised manuscript (line68-72). Thank you.
- Table 5 was added in response to question about experimentally confirmed epitopes. However, this table should also contain information on HLA alleles and refer to studies where each epitope was investigated.
Answer: Thank you for your valuable suggestion. We have added the HLA alleles information and the corresponding references to the revised Table 5 (page 10).
Table 5. The epitopes verified by experiment contained in mosaic sequence
|
Protein name |
Epitope verified by experiment |
CD4+/CD8+ T cell response |
HLA restriction(s) |
position |
Mosaic Sequence |
|
|
S protein |
EYVSQPFLM [22] |
CD8+ |
A*11:01; C*07:02 |
169-177 |
Seq2 |
|
|
KSTNLVKNK [22] |
CD8+ |
A*02:01; B*40:01 |
529-537 |
Seq2 |
||
|
N protein |
QRNAPRITF [22] |
CD8+ |
A*11:01; B*40:01; C*07:02 |
9-17 |
Seq3 |
|
|
SPRWYFYYL [22] |
CD8+ |
A*02:01; C*07:02 |
105-113 |
Seq3 |
||
|
YLGTGPEAGL [23] |
CD8+ |
A*02:01; C*07:02 |
112-121 |
Seq3 |
||
|
KTFPPTEPK [22] |
CD8+ |
A*11:01; C*07:02 |
361-369 |
Seq3 |
||
|
M protein |
PKEITVATSRTLSYYKL [22] |
CD4+ |
DRB1*07:01; |
164-180 |
Seq1 |
|
|
FLWLLWPVTL [23] |
CD8+ |
A*02:01 |
53-62 |
Seq4 |
||
|
FLFLTWICL [23] |
CD8+ |
A*02:01 |
26-34 |
Seq4 |
||
|
KLLEQWNL [23] |
CD8+ |
A*02:01 |
15-22 |
Seq4 |
||
|
E protein |
FLLVTLAIL [23] |
CD8+ |
A*02:01 |
26-34 |
Seq4 |
|
|
SEETGTLIVNSVLLF [24] |
CD4+ |
DQA1*0501;DQB1*0301 |
6-20 |
Seq4 |
||
- Supplementary files should have their names and be indicated at the end of the manuscript (lanes 304-305).
Answer: Thank you for your kind reminders. We have added this information at the end of our revised manuscript as you suggested.
Reviewer 2 Report
The authors provided satisfactory responses to my inquires
Author Response
The authors provided satisfactory responses to my inquires
Answer: Thank you very much for your kind approval of our work.